# Sustaining a temperature difference

**Matteo Polettini and Alberto Garilli**⋆

Physics and Materials Science Research Unit, University of Luxembourg, Campus
Limpertsberg, 162a avenue de la Faïencerie, L-1511 Luxembourg (G. D. Luxembourg)

⋆ alberto.garilli@uni.lu

## Abstract

We derive an expression for the minimal rate of entropy that sustains two reservoirs at different temperatures $T_0$ and $T_\ell$. The law displays an intuitive $\ell^{-1}$ dependency on the relative distance and a characterisic $\log^2(T_\ell/T_0)$ dependency on the boundary temperatures. First we give a back-of-envelope argument based on the Fourier Law (FL) of conduction, showing that the least-dissipation profile is exponential. Then we revisit a model of a chain of oscillators, each coupled to a heat reservoir. In the limit of large damping we reobtain the exponential and squared-log behaviors, providing a self-consistent derivation of the FL. For small damping "equipartition frustration" leads to a well-known ballistic behaviour, whose incompatibility with the FL posed a long-time challenge.

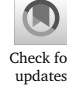

# 1   Introduction

Temperature differences and gradients are a common motif in the physics of systems out of equilibrium. Primarily they serve as fixed boundary conditions for studying how energy flows *within* a system [1–3] and couples to other currents, e.g. electric or matter ones [4–6]. Here instead we will be interested in a dual question: what is the least amount of energy that has to be dissipated *outside* the system by some apparatus whose only task is to sustain a temperature difference?

More precisely, in this work we study lower bounds to the entropy production rate (EPR) $\sigma$ of a conductor with respect to the profile of temperatures in the bulk, given those at the boundary. In linear systems held at temperatures $T_0$ and $T_\ell$ at the extremities we find the simple expression $\sigma^* \propto \ell^{-1} \log^2(T_\ell/T_0)$. We first provide a simple heuristic argument based on the Fourier Law (FL) of conduction, and then rederive our results in a stochastic model of a linear chain of harmonic oscillators coupled to heat reservoirs, analyzed in the light of the First and Second laws of (stochastic) thermodynamics [7]. As a main new technical result we obtain explicit second-order expressions for the stationary distribution and the EPR, that may be applied in the optimization of more complex networks of interacting nodes at different temperatures (e.g. power grids [8–10]).

We also approach some foundational issues from a new angle. Microscopic derivations of the FL have for long been considered a challenge to the theorist [2, 11–13]. The problem here is to reconcile the ballistic behaviour in the bulk, that is supposed to be adiabatically isolated from the environment, and the diffusive character of heat conduction. In our approach, along similar lines as in Refs. [14–18], we open the bulk to interactions with the environment. For example, if we think of a refrigerator with $T_0$ and $T_\ell$ respectively the temperatures inside and

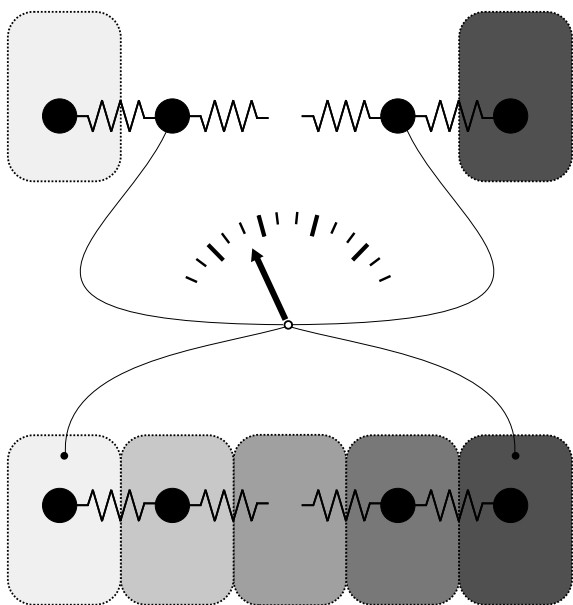

Figure 1: Illustration of *constructive* vs. *self-consistent* approaches to heat conduction in temperature gradients. In the former heat flows from the hot reservoir to the cold one through the oscillators. It former draws its motivation in the derivation of the physics of open systems from that of isolated systems (micro-to-meso/scopic); its dissipative observables are the temperatures of the bulk oscillators. In the latter heat flows to and from each reservoir. This approach focuses on the temperature of the baths, and aims to connect to broad-scale energy consumption (meso-to-macro/scopic).

outside the cold room, in the "challenge" $T(x), x \in (0, \ell)$ would represent the temperature profile within the adiabatic walls, while in our approach it rather describes the temperature of the refrigerating liquid in the cooling coil (see Fig. 1). In the so-called overdamping limit, the FL emerges self-consistently (but not constructively), while the low-noise limit leads to the known ballistic behaviour, and to a phenomenon of "frustration" whereby the system's bulk temperatures differ from their environment's. Furthermore, notice that for fixed $T_\ell$, letting $T_0 \to 0$ the EPR diverges: reaching zero temperature requires ever-increasing dissipated power, providing a self-consistent formulation of the Third Law of thermodynamics – that thus is not independent of the First and Second.

## 2 Heuristics

We consider an extended system whose degrees of freedom are localized by position $x \in X$ in space, and for which it makes sense to talk about a local temperature $T(x)$, constrained by some boundary values $T(\partial X)$. The existence of a meaningful temperature is usually called *local equilibrium* or *local detailed balance*. We point out that it is not the system itself to be at equilibrium, but rather a continuous set of thermometers whose local degrees of freedom are labeled by $x$, and that interact via the system.

To a temperature gradient is associated a thermal force $\vec{F} = \vec{\nabla} T^{-1}$, which generates a heat current $\vec{J}$. If we provisionally take the FL for granted, the stationary heat current is given by $\vec{J} = -\kappa \vec{\nabla} T$, with $\kappa$ the heat conductivity. The macroscopic stationary EPR is given by the scalar product

$$\sigma := \int_X \vec{F} \cdot \vec{J} \tag{1}$$

$$= \kappa \int_X |\nabla \log T|^2 \tag{2}$$

$$= \kappa \int_X \frac{1}{T} \Delta T - \kappa \int_{\partial X} \vec{n} \cdot \vec{\nabla} \log T . \tag{3}$$

We rewrote the expression in a couple of equivalent useful ways. where $\vec{n}$ is the unit vector orthogonal to the boundary surface element. Notice that the second expression in Eq. (1) manifests an invariance under the transformation $T \to 1/T$.

We search for extremals of $\sigma$ by taking the functional derivative $\delta/\delta T(y)$. After some standard integration by parts and application of Dirac deltas (see details in the Appendix ) we obtain the Laplace equation $\Delta \log T^* = 0$, where the asterisk stands for the extremal. Plugging into the EPR we find

$$\sigma^* = \frac{\kappa}{2} \int_{\partial X} \vec{n} \cdot \vec{\nabla} \log^2 T^*, \tag{4}$$

which is easily checked to be a minimum. Notice the squared-log dependency on boundary temperatures.

We now reduce to one dimension by assuming that the temperature only varies in one extended direction $x \in [0, \ell]$, while at fixed $x$ it is uniform in a perpendicular area of fixed size $\alpha$. Subjecting the Laplace equation to the boundary constraints $T(0) = T_0$ and $T(\ell) = T_\ell$, we obtain as the unique minimum profile

$$T^*(x) = T_0 \left( \frac{T_\ell}{T_0} \right)^{x/\ell} . \tag{5}$$

The least EPR is then given by

$$\sigma^* = \frac{\kappa\alpha}{\ell}\log^2(T_\ell/T_0). \tag{6}$$

Notice that the shorter the distance, the steeper the gradient, the higher the EPR.

## 3 Model

Let us now re-derive the above results in a well-known microscopic model, employed e.g. in Refs. [17,18] in attempts at derivations of the FL by self-consistent reservoirs, and in Ref. [16] to discuss energy equipartition of normal modes.

We consider a homogeneous chain of $n$ harmonic oscillators of unit mass placed at regularly spaced positions $x_k = (k-1)\ell/(n-1)$, with $k = 1,\ldots,n$, between boundary $x = 0$ and $x = \ell$. We also set Boltzmann's constant to unity $k_B = 1$. We denote $q_k$ the amplitude of oscillation of the $k$-th oscillator and $p_k$ its momentum, and collect $\boldsymbol{z} = (q_k, p_k)_{k=1}^n$. These amplitues are the effective degrees of freedom that describe the local interaction of the system with the baths. The total energy is $H(\boldsymbol{z}) = \frac{1}{2}\sum_{k=1}^{n+1}\left[p_k^2 + \omega^2(q_k - q_{k-1})^2\right]$, where $q_0 = p_0 = q_{n+1} = p_{n+1} = 0$ stand for the non-oscillating endpoints where the chain is anchored and $\omega$ is the angular frequency, which for the moment we also set to unity $\omega = 1$ to resume it later in the discussion. Each harmonic oscillator is in contact with a heat bath at temperature $T_k$, which is a source of stochastic white noise (the reduced effect of the bath degrees of freedom). The dynamics is described by the Langevin equation

$$\dot{q}_k(t) = \frac{\partial}{\partial p_k}H(\boldsymbol{z}(t)) \tag{7a}$$

$$\dot{p}_k(t) = -\frac{\partial}{\partial q_k}H(\boldsymbol{z}(t)) - \gamma p_k + \sqrt{\gamma T_k}\,\zeta_k(t), \tag{7b}$$

where $\zeta_k(t)$ are the formal time derivatives of independent Brownian motions, and $\gamma$ is the damping coefficient. Letting $D$ be the diagonal positive-definite diffusion matrix $D = \text{diag}\{T_k\}_{k=1}^n$, and $A$ the symmetric tridiagonal matrix

$$A = \begin{pmatrix} 2 & -1 & & & \\ -1 & 2 & -1 & & \\ & -1 & \ddots & \ddots & \\ & & \ddots & & -1 \\ & & & -1 & 2 \end{pmatrix}, \tag{8}$$

and further defining the $2n \times 2n$ matrices

$$\mathbb{M} = \begin{pmatrix} 0 & -I \\ A & \gamma I \end{pmatrix}, \qquad \mathbb{D} = \begin{pmatrix} 0 & 0 \\ 0 & \gamma D \end{pmatrix}, \tag{9}$$

with $I$ the identity, the equations of motion take the form of a multidimensional Ornstein-Uhlenbeck (OU) process

$$\dot{\boldsymbol{z}} = -\mathbb{M}\boldsymbol{z} + \sqrt{\mathbb{D}}\,\zeta, \tag{10}$$

where $\zeta(t)$ are $2n$ independent delta-correlated white noises. Letting $\rho(\boldsymbol{z},t)d\boldsymbol{z}$ be the probability of $\boldsymbol{z}(t)$ being in a neighborhood of $\boldsymbol{z}$, its density satisfies the Kramers diffusion equation

$$\frac{\partial}{\partial t}\rho = \frac{\partial}{\partial \boldsymbol{z}}\cdot\left(\mathbb{M}\boldsymbol{z}\rho + \mathbb{D}\frac{\partial}{\partial \boldsymbol{z}}\rho\right) \tag{11}$$

$$= \{H,\rho\} - \frac{\partial}{\partial \boldsymbol{p}}\cdot\boldsymbol{j}. \tag{12}$$

In the second identity in Eq. (12), $\{\cdot,\cdot\}$ denotes the Poisson bracket and the dissipative current is given by [19, 21]

$$\boldsymbol{j} := -\gamma\left(\boldsymbol{p} + D\frac{\partial}{\partial \boldsymbol{p}}\right)\rho, \tag{13}$$

showing that the underdamped dynamics clearly separates into a ballistic term and a diffusive one.

The stationary probability $\rho_\infty(\boldsymbol{z}) := \lim_{t\to\infty}\rho(\boldsymbol{z},t)$ is found by assuming as *ansatz* a centered multinormal distribution $\rho_\infty(\boldsymbol{z}) \propto \exp-\frac{1}{2}\boldsymbol{z}\cdot\mathbb{C}^{-1}\boldsymbol{z}$ where

$$\mathbb{C} = \begin{pmatrix} C_{qq} & C_{pq} \\ C_{pq}^{\mathsf{T}} & C_{pp} \end{pmatrix} \tag{14}$$

is the stationary correlation matrix, with $C_{qq}$ and $C_{pp}$ symmetric, and $^{\mathsf{T}}$ denoting transposition.

# 4 Results

## 4.1 Stationary distribution

For the sake of generalization we momentarily allow for a non-symmetric $A$. Plugging the stationary distribution into the continuity equation $\partial_{\boldsymbol{p}}\cdot\boldsymbol{j}_\infty = 0$ one finds that the covariance matrix is uniquely determined by the (first-order) Lyapunov equation [22, 23]

$$\mathbb{C}\mathbb{M}^{\mathsf{T}} + \mathbb{M}\mathbb{C} = 2\mathbb{D}. \tag{15}$$

In view of Eq. (14), one finds that $C_{pq}^{\mathsf{T}} = -C_{pq}$ is antisymmetric, and furthermore:

$$AC_{qq} - C_{qq}A^{\mathsf{T}} = 2\gamma C_{pq} \tag{16a}$$

$$C_{qq}A^{\mathsf{T}} + AC_{qq} = 2C_{pp} \tag{16b}$$

$$AC_{pq} - C_{pq}A^{\mathsf{T}} = 2\gamma(D - C_{pp}). \tag{16c}$$

The first defines $C_{qq}$ in terms of $C_{pq}$, the second $C_{qq}$ in terms of $C_{pp}$, and the third $C_{pq}$ in terms of $C_{pp}$. Combining these formulas together we find the *second order Lyapunov equation* for the displacements' covariance $C := C_{qq}$

$$\frac{A^2 C - 2ACA^{\mathsf{T}} + CA^{\mathsf{T}2}}{2\gamma^2} + AC + CA^{\mathsf{T}} = 2D. \tag{17}$$

In the overdamping limit $\gamma \to \infty$ this expression reduces to the well-known $AC + CA^{\mathsf{T}} = 2D$ [22–24]; furthermore we have $C_{pp} \to D$ (local equipartition) and $C_{pq} \sim \frac{1}{\gamma}(AC - D)$. Defining the antisymmetric "curvature" tensor $R := D^{-1}A - A^{\mathsf{T}}D^{-1} = 0$ [20], the condition of (global) detailed balance states that $R$ vanishes, if and only if $C_{pq}$ vanishes [25]. *Proof.* If detailed balance holds then $C = DA^{\mathsf{T}-1}$ solves the second order Lyapunov equation and $C_{pq} = 0$. Vice versa, if $C_{pq} = 0$ then the first term in Eq. (17) vanishes, one obtains $D = AC = CA^{\mathsf{T}}$ and therefore $R = 0$. $\square$ In our case detailed balance is achieved when $D \propto I$, i.e. for all equal temperatures.

## 4.2 Entropy production rate

Before turning to the solution of the second-order Lyapunov equation, we introduce thermodynamic quantities. We now take the time derivative of the mean energy

$$\frac{d}{dt}\langle H(\boldsymbol{z}(t))\rangle = -\gamma\left(\langle|\boldsymbol{p}(t)|^2\rangle - \operatorname{tr}D\right), \tag{18}$$

where the average is over realizations of the Brownian motions, and we used the Itô Lemma and the martingale property. The First Law $d\langle H \rangle + \sum_k đQ_k = 0$ suggests to define the heat flow from the $k$-th reservoir as

$$\frac{đQ_k}{dt}(t) := \gamma\left(\langle p_k^2(t)\rangle - T_k\right).$$ (19)

Notice that it vanishes when equipartition holds. The Clausius formula

$$\sigma := \frac{dS}{dt} + \sum_k \frac{1}{T_k}\frac{đQ_k}{dt} = \int d\boldsymbol{z}\, \boldsymbol{j}(\gamma\rho D)^{-1}\boldsymbol{j}$$ (20)

defines the EPR, where $S := -\int d\boldsymbol{z}\,\rho\log\rho$ is the Gibbs-Shannon entropy. The second identity, providing the Second Law $\sigma \geq 0$, is proven in Appendix B. At the stationary state, plugging the definition of the heat flux Eq. (19) and the expression for the diffusion matrix we find $\sigma_\infty = \gamma\,\mathrm{tr}\left[D^{-1}(C_{pp} - D)\right]$. We can now employ Eq. (16c) to obtain $\sigma_\infty = -\mathrm{tr}(C_{pq}R)/2$, and finally employing the Lyapunov equation for $C_{pq}$ and its antisymmetry Eq. (16a) we arrive at

$$\sigma_\infty = \frac{1}{2\gamma}\,\mathrm{tr}\left[(C_{qq}A^\mathsf{T} - AC_{qq})D^{-1}A\right].$$ (21)

Both this latter expression for the EPR and the Lyapunov equation are clearly independent of the vector basis chosen to represent matrices. We will now work with normal modes, i.e. orthonormal eigenvectors $\boldsymbol{a}_\alpha$ of $A$. Then $A = \sum_\alpha \lambda_\alpha \boldsymbol{a}_\alpha \otimes \boldsymbol{a}_\alpha$, where $\lambda_\alpha$ are the real eigenvalues and $\otimes$ denotes the outer product. In this basis the diffusion and covariance matrices have entries respectively $D_{\alpha\beta} = \boldsymbol{a}_\alpha \cdot D\boldsymbol{a}_\beta$ and $C_{\alpha\beta} = \boldsymbol{a}_\alpha \cdot C\boldsymbol{a}_\beta$. Finally we can express Eq. (17) in this basis to find $C_{\alpha\beta} = 2D_{\alpha\beta}/[\lambda_\alpha + \lambda_\beta + (\lambda_\alpha - \lambda_\beta)^2/2\gamma^2]$ and

$$\sigma_\infty = -\gamma\sum_{\alpha,\beta} D_{\alpha\beta}(D^{-1})_{\alpha\beta}\frac{(\lambda_\alpha - \lambda_\beta)^2}{(\lambda_\alpha - \lambda_\beta)^2 + 2(\gamma/\omega)^2(\lambda_\alpha + \lambda_\beta)},$$ (22)

where we finally resumed the angular frequency $\omega$ simply by rescaling all eigenvalues $\lambda_\alpha \to \omega^2\lambda_\alpha$.

In our problem, the tridiagonal matrix $A$ has real eigenvalues $\lambda_\alpha = 2 + 2\cos(\alpha\pi/(n+1))$ and orthonormal eigenvectors' entries $a_\alpha^k = \sqrt{2/(n+1)}\sin(\alpha k\pi/(n+1))$, for $\alpha, k = 1, \ldots, n$, ranging from slower to faster modes (see Appendix C and Ref. [26]). We can thus compute $D_{\alpha\beta}$; notice that by orthonormality, $(D^{-1})_{\alpha\beta}$ can be obtained from $D_{\alpha\beta}$ by the duality transformation $T_k \to 1/T_k$. Finally we obtain

$$\sigma = -\gamma\sum_{k,h}\frac{\Delta_{k,h}T_h}{T_k}$$ (23)

$$= \frac{\gamma}{2}\sum_{k<h}\left(\frac{1}{T_k} - \frac{1}{T_h}\right)\Delta_{kh}(T_k - T_h)$$ (24)

$$= \sum_{k\neq h}F_{kh}J_{kh},$$ (25)

where in the first expression

$$\Delta_{kh} = \sum_{\alpha,\beta}a_\alpha^k a_\beta^k a_\alpha^h a_\beta^h\frac{(\lambda_\alpha - \lambda_\beta)^2}{(\lambda_\alpha - \lambda_\beta)^2 + 2(\gamma/\omega)^2(\lambda_\alpha + \lambda_\beta)}.$$ (26)

Matrix $\Delta$ is symmetric, its diagonal entries are positive, and by orthonormality $\sum_k a_\alpha^k a_\beta^k = \delta_{\alpha,\beta}$ one finds that $\sum_k \Delta_{kh} = 0$. Therefore $\Delta$ is a proper discretized Laplacian (see also Refs.

[27, 28] in the quantum case), and Eq. (23) is a discretized equivalent of Eq. (3) without the boundary term because in the discrete case this is directly absorbed in the definition of the Laplacian. Eq. (1) is instead reminiscent of Eq. (24) and Eq. (25) with $F_{k,h} := 1/T_k - 1/T_h$ as the thermodynamic force due to the competition of the $k$-th and the $h$-th reservoir and $J_{k,h} := \gamma \Delta_{kh}(T_k - T_h)$, reproducing the traditional bilinear structure of the entropy production rate [29]. Then the main difference with the heuristic case is that, despite the fact that the oscillators only interact with nearest neighbors, they create all-to-all nonequilibrium currents (see also Ref. [30]). However, Fig. 2 shows that for large $\gamma$ first-neighbor contributions indeed dominate.

Finally, we minimize the EPR with respect to the temperatures of the bulk oscillators, subject to constrained values of the temperatures of the first and last oscillators. For $n = 3$ the free oscillator's temperature is easily found to be $T_2^* = \sqrt{T_1 T_3}$ (see Appendix D), yielding an analytical expression of the minimum EPR whose most interesting feature is that as a function of $\gamma$ it vanishes for $\gamma \to 0, \infty$ and has a maximum in between, whose physical significance is still an open question.

For $n > 1$ minimization could only be achieved computationally. Fig. 3 shows that in the overdamping limit in the oscillator model the optimal temperature profile is consistent with that from the FL, while in the underdamped limit the bulk oscillators's optimal temperatures flatten, reproducing the behaviour observed in Ref. [14]. With crosses we plotted the mean squared momentum of the oscillators, calculated as

$$\langle p_k^2 \rangle = \sum_{\alpha, \beta} \frac{D_{\alpha\beta} a_k^\alpha a_k^\beta}{1 + \frac{\omega^2(\lambda_\alpha - \lambda_\beta)^2}{2\gamma^2(\lambda_\alpha + \lambda_\beta)}}. \tag{27}$$

While in the overdamping limit equipartition is reached, in the low-damping limit a phenomenon of *temperature frustration* occurs, whereby the bulk's internal temperatures are slightly off the environment's. A similar self-consistent treatment of this system was given in Ref. [15]; there, instead of minimizing the EPR, the Authors looked for the temperature profile that more

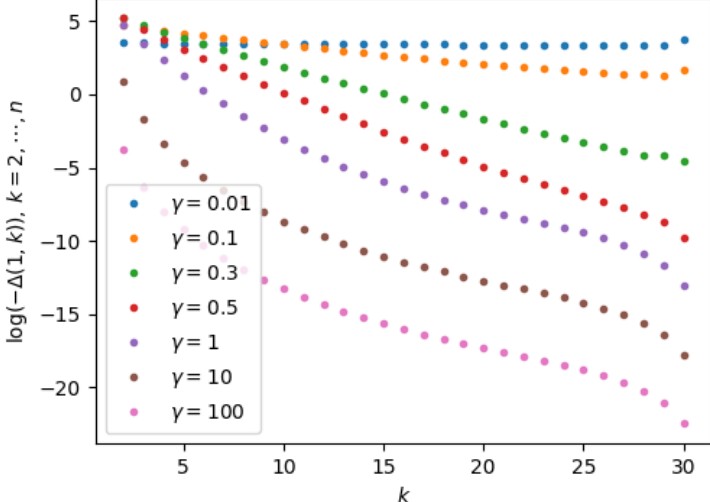

Figure 2: Plot of $-\Delta_{1,k}$ in log scale for a chain of $n = 30$ oscillators, for different values of $\gamma$. In the overdamping limit first neighbours are strongly favoured, while in the underdamped limit spatial proximity in the bulk does not play a role, and only the endpoint oscillators are slightly favoured.

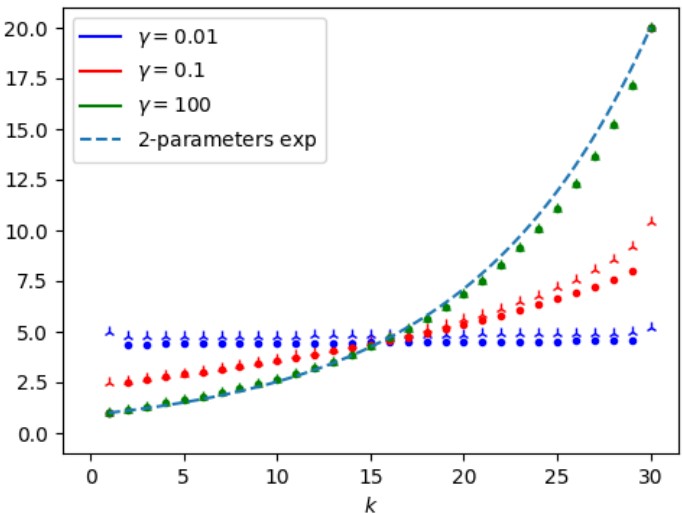

Figure 3: For a chain 30 oscillators with fixed boundary temperatures $T_1 = 1$ and $T_{30} = 20$, and for different values of $\gamma = 0.01, 0.1, 100$, in bullets the bulk temperature profile that minimizes the EPR, and in crosses the oscillators' mean-squared momentum. The fitting curve is an exponential with fixed endpoints, as in Eq. (5).

closely satisfies equipartition over all of the chain, finding that it is linear, instead of exponential. This difference can be explained by the fact that to minimize the EPR it is best to reduce the heat flow to the colder reservoirs. This explains why e.g. in the red curve in Fig. 3 departure from equipartition occurs at hotter temperatures.

Finally, in the overdamping limit the overlapping of all curves in Fig. 4 proves the squared-log behaviour and the fact that, for given temperature difference well beyond the linear regime, the minimal EPR scales like $(n+1)^{-1}$, reproducing the $\ell^{-1}$ dependence in the FL.

# 5   Possible developments

To conclude, in this paper we collected theoretical evidence that, for systems open to the interaction with the environment, on the assumption that thermometers can be defined locally, the minimal entropic cost of maintaining a temperature gradient at the two extremities of a linearly extended body scales with the squared logarithm of the temperature ratio, and inversely with the distance. A main novelty of this work is the idea of viewing a temperature gradient "from the outside" instead of "from the inside". This may be of practical interest in the assessment of the industrial scaling of the energetic demand of technologies running at very low temperatures or in the optimization of power grids. Our law may provide an indirect testing ground for the thermodynamics of open systems based on Markov processes, whose experimental verification is still intertwined with the identification of the kind systems to which it applies. Finally, notice that experimental verification of the squared-log behaviour would entail a form of "minimum entropy production" principle [31].

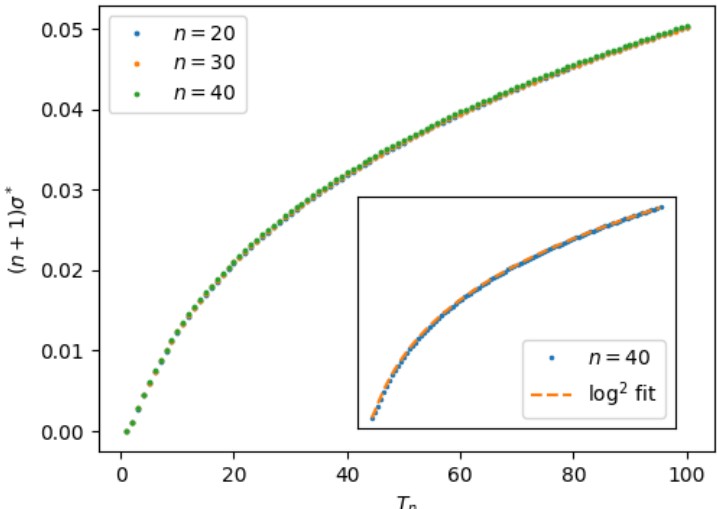

Figure 4: In the overdamped case $\gamma = 100$, for chains of $n = 20, 30, 40$ oscillators, $n + 1$ times the minimum EPR $\sigma^*$ as a function of the temperature of the right-most oscillator ranging from 1 to 100, for given $T_0 = 1$. The three data sets overlap and are fitted by a squared-log curve as in Eq. (6) (see example for $n = 40$ in the subplot).

## 6 Funding information

The research was supported by the National Research Fund Luxembourg (project CORE ThermoComp R-AGR-3425-10) and by the European Research Council, project NanoThermo (ERC-2015-CoG Agreement No. 681456).

## A Minimal EPR

We want to explicitly show that the temperature profile $T^*$ that minimizes the macroscopic stationary EPR with our notion of local temperature follows the Laplace equation

$$\Delta \log T^* = 0. \tag{A.1}$$

By considering ($\kappa = 1$)

$$\sigma = \int_X |\nabla \log T|^2 = \int_X \left(\frac{\nabla T}{T}\right)^2 \tag{A.2}$$

the entropy $\sigma = \sigma[T]$ is a functional of the temperature field $T(x)$. By taking the functional derivative of $\sigma$ respect to $T$

$$\frac{\delta \sigma[T]}{\delta T(x)} = \frac{\partial}{\partial T}\left[\left(\frac{\nabla T}{T}\right)^2\right] - \nabla \cdot \frac{\partial}{\partial(\nabla T)}\left[\left(\frac{\nabla T}{T}\right)^2\right] = 0,$$

where we applied Dirac's deltas coming from the functional derivative. The first term is

$$\frac{\partial}{\partial T}\left[\left(\frac{\nabla T}{T}\right)^2\right] = 2\left(\frac{\nabla T}{T}\right)\cdot\left(-\frac{\nabla T}{T^2}\right) = -\frac{2}{T}(\nabla \log T)^2$$

and the second one gives

$$\nabla \cdot \left( 2 \left( \frac{\nabla T}{T} \right) \frac{\partial}{\partial \nabla T} \left( \frac{\nabla T}{T} \right) \right) = 2 \nabla \cdot \left( \frac{1}{T} \frac{\nabla T}{T} \right)$$

$$= 2 \nabla \cdot \left( \frac{1}{T} \nabla \log T \right) = -\frac{2}{T} (\nabla \log T)^2 + \frac{2}{T} \Delta \log T ,$$

then

$$\frac{\delta \sigma[T]}{\delta T(x)} = -\frac{2}{T} (\nabla \log T)^2 + \frac{2}{T} (\nabla \log T)^2 - \frac{2}{T} \Delta \log T = 0 ,$$

which is true when $\Delta \log T^* = 0$ holds. The minimal EPR is obtained by integrating (A.2) by parts:

$$\int_X |\nabla \log T^*|^2 = \int_X \nabla \cdot (\log T^* \nabla \log T^*) - \int_X \log T^* \cdot \Delta \log T^* .$$

The second contribution vanishes because of Eq. (A.1), then

$$\sigma = \int_X \nabla \cdot (\log T^* \nabla \log T^*) = \frac{1}{2} \int_X \nabla \cdot \nabla \log^2 T^*$$

$$= \frac{1}{2} \int_{\partial X} n \cdot \nabla \log^2 T^* .$$

When restricting our study to one dimension the Laplace equation gives the minimal exponential profile

$$T(x) = T_0 \left( \frac{T_\ell}{T_0} \right)^{x/\ell} , \tag{A.3}$$

since $\partial_x T((1/T)\partial_x T) = 0$ when $\partial_x T = \text{const } T$ which is a first-order differential equation whose solution is a 2-parameters exponential. The parameters are obtained with the two conditions $T(0) = T_0$ and $T(\ell) = T_\ell$.

# B   Underdamped EPR

Consider the second expression in Eq. (14). We have

$$k_B \int d\boldsymbol{z} \boldsymbol{j} (\gamma \rho D)^{-1} \boldsymbol{j} \tag{B.1}$$

$$= \gamma k_B \int d\boldsymbol{z} \left( \boldsymbol{p}\rho + D \frac{\partial \rho}{\partial \boldsymbol{p}} \right) \cdot \left( D^{-1} \boldsymbol{p} + \frac{\partial}{\partial \boldsymbol{p}} \log \rho \right)$$

$$= \gamma k_B \int d\boldsymbol{z} \left( \boldsymbol{p} \cdot D^{-1} \boldsymbol{p} + \boldsymbol{p} \cdot \frac{\partial}{\partial \boldsymbol{p}} \right) \rho + k_B \int d\boldsymbol{z} \log \rho \frac{\partial}{\partial \boldsymbol{p}} \cdot \boldsymbol{j} , \tag{B.2}$$

where in the second term we integrated by parts and recovered the definition of the current. Now, employing the continuity equation Eq. (8) we obtain for the second term

$$k_B \int d\boldsymbol{z} \log \rho \frac{\partial}{\partial \boldsymbol{p}} \cdot \boldsymbol{j} = k_B \int d\boldsymbol{z} \log \rho \left( \{H, \rho\} - \frac{\partial}{\partial t} \rho \right) = \frac{d}{dt} S(t) , \tag{B.3}$$

where $S(t) = -k_B \int z \rho(z,t) \log \rho(z,t)$ is the Gibbs-Shannon entropy, and we used the Liouville theorem and probability conservation. Also, notice that $\int dz\, p \cdot D^{-1} p\, \rho = \mathrm{tr}[D^{-1} C_{pp}(t)]$ leading to

$$k_B \int dz\, j (\gamma \rho D)^{-1} j \tag{B.4}$$

$$= \frac{d}{dt} S(t) + \gamma k_B \,\mathrm{tr}[D^{-1} C_{pp}(t)] + \gamma k_B \int dz\, p \cdot \frac{\partial}{\partial p} \rho(z,t). \tag{B.7}$$

Finally this last term can be integrated by parts yielding $-n\gamma k_B$, thus recovering the first expression in Eq. (20).

## C  Orthonormality of the eigenvectors of $A$

We want to show that the basis of eigenvectors $\{a_\alpha\}$, $\alpha = 1,\dots,n$ with components $a_\alpha^k = \sqrt{\frac{2}{n+1}} \sin\left(\frac{\alpha k \pi}{n+1}\right)$ is orthonormal. The condition

$$\sum_{k=1}^{n} a_\alpha^k a_\alpha^k = 1 \tag{C.1}$$

is fulfilled since

$$\sum_{k=1}^{n} \sin^2\left(\frac{\alpha k \pi}{n+1}\right) = -\frac{1}{4} \sum_{k=1}^{n} \left(-2 + e^{\frac{2i\alpha k\pi}{n+1}} + e^{-\frac{2i\alpha k\pi}{n+1}}\right)$$

$$= \frac{n}{2} - \frac{1}{4}\left(\sum_{k=1}^{n}\left(e^{\frac{2i\alpha\pi}{n+1}}\right)^k + \sum_{k=1}^{n}\left(e^{-\frac{2i\alpha\pi}{n+1}}\right)^k\right) = \frac{n+1}{2}, \tag{C.2}$$

where we used that $\sin(x) = \frac{1}{2i}(e^{ix} - e^{-ix})$ and that the partial sum of the geometric series is given by $\sum_{k=0}^{m} x^k = (1 - x^{m+1})/(1-x)$. The result is obtained by noticing that our summations runs from $k = 1$ and that $e^{2iz\pi} = \cos(2z\pi) = 1, \forall z \in \mathbb{Z}$.

With similar arguments we can prove that $\{a_\alpha\}$ is also orthogonal: in fact

$$\sum_{k=1}^{n} a_\alpha^k a_\beta^k = -\frac{2}{(n+1)} \frac{1}{4} \sum_{k=1}^{n} \left(e^{i\alpha k\pi/(n+1)} - e^{-i\alpha k\pi/(n+1)}\right)\left(e^{i\beta k\pi/(n+1)} - e^{-i\beta k\pi/(n+1)}\right)$$

$$= -\frac{2}{(n+1)}\frac{1}{4}\left[\frac{1 - e^{i(\alpha+\beta)\pi}}{1 - e^{i(\alpha+\beta)\pi/(n+1)}} - \frac{1 - e^{i(\alpha-\beta)\pi}}{1 - e^{i(\alpha-\beta)\pi/(n+1)}} - \frac{1 - e^{-i(\alpha-\beta)\pi}}{1 - e^{-i(\alpha-\beta)\pi/(n+1)}} + \frac{1 - e^{-i(\alpha+\beta)\pi}}{1 - e^{-i(\alpha+\beta)\pi/(n+1)}}\right]$$

$$= \delta_{\alpha,\beta}, \tag{C.3}$$

since for $\alpha = \beta$ we fall in the previous case to show normality of $a_\alpha$, and for $\alpha \neq \beta$ we can say that when $\alpha + \beta$ is even (odd) also $\alpha - \beta$ is even (odd), and in both case the expression above vanishes because $\cos(m\pi) = 1$ for even $m$ and $\cos(m\pi) = -1$ for odd $m$.

## D  Exact solution for 3 oscillators

Let us consider a system of 3 interacting harmonic oscillators where the endpoints (oscillators 1 and 3) are coupled with thermal reservoirs with given temperatures $T_1 = T_c$ and $T_3 = T_h$.

We want to find the temperature $T$ of the oscillators in the middle which minimizes the EPR given by Eq. (17)

$$\sigma = \gamma \sum_{k<h} \left( \frac{1}{T_h} - \frac{1}{T_k} \right) \Delta_{kh} (T_h - T_k) \quad h = 1, 2, 3. \tag{D.1}$$

The expression above can be written explicitly as

$$\sigma = \gamma \left[ \left( \frac{1}{T_h} - \frac{1}{T_c} \right) \Delta_{13}(T_h - T_c) + \left( \frac{1}{T_h} - \frac{1}{T} \right) \Delta_{23}(T_h - T) + \left( \frac{1}{T} - \frac{1}{T_c} \right) \Delta_{12}(T - T_c) \right] \tag{D.2}$$

and can be minimized by taking the derivative respect to $T$ obtaining

$$\frac{d\sigma}{dT} = \gamma \frac{d}{dT} \left( -\frac{T_h}{T} \Delta_{23} - \frac{T}{T_h} \Delta_{23} - \frac{T}{T_c} \Delta_{12} - \frac{T_c}{T} \Delta_{12} \right) = 0. \tag{D.3}$$

The elements $\Delta_{kh}$ with $k, h = 1, 2, 3$ include the dependence on the damping parameter $\gamma$, since in the general underdamped framework (see Eq. (26))

$$\Delta_{kh} = \sum_{\alpha,\beta=1}^{3} a_\alpha^k a_\beta^k a_\alpha^h a_\beta^h \frac{(\lambda_\alpha - \lambda_\beta)^2}{(\lambda_\alpha - \lambda_\beta)^2 + 2(\gamma/\omega)^2 (\lambda_\alpha + \lambda_\beta)}, \tag{D.4}$$

with $\lambda_\alpha = 2 + 2\cos\left(\frac{\alpha\pi}{4}\right)$, $\alpha = 1, 2, 3$ denoting the $\alpha$-th eigenvalue of $A$ and $a_\alpha^k = \frac{1}{\sqrt{2}} \sin\left(\frac{\alpha k\pi}{4}\right)$ the $k$-th component, $k = 1, 2, 3$ of the eigenvector $a_\alpha$ associated to $\lambda_\alpha$.

The matrix $\Delta$ is symmetric respect to the diagonal, but it is also symmetric respect to the anti-diagonal. This means that $\Delta_{12} = \Delta_{23}$. Then the minimization above gives

$$\frac{1}{T^2}(T_h + T_c) = \frac{1}{T_h} + \frac{1}{T_c} \quad \Rightarrow \quad T = \sqrt{T_c T_h} \tag{D.7}$$

and we lost any dependence on $\gamma$ for the optimal temperature $T$. Then the temperature profile for 3 oscillators is independent on $\gamma$ and is always exponential. If we consider now a larger system with a generic number $n$ of oscillators, with given temperatures at the endpoints, we obtained in the overdamped limit that the optimal temperature profile is exponential from $T_c$ to $T_h$. In the underdamped situation the profile is quite different: by taking $\gamma$ smaller and smaller, the temperature profile of all the oscillators in the bulk of the system (excluding then the endpoints that are still at the given temperatures $T_c$ and $T_h$) tends to be uniform with a value $T = \sqrt{T_c T_h}$ as if the oscillators in the bulk are behaving collectively as a single oscillator with temperature $T$. This also gives that the minimal profile depends on $\gamma$, which is natural since the explicit minimization (analytically impossible) would contain explicit dependence on the elements of $\Delta$ and in the limit $\gamma \to 0$ we lose any dependence on $\gamma$ as in the 3 oscillators model.

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
