# Peer review of "Sustaining a temperature difference"

_SciPost Physics, doi:SciPost Phys. 9, 030 (2020)_

## Round 1 · Referee Report · Anonymous (Referee 1) · 2020-7-2

Report

This paper is clear and enjoyable to read, and I fully recommend publication once some minor points are addressed.

  1. In the introduction the authors state "As a main new technical result we obtain explicit second-order expressions for the stationary distribution and the EPR". The new result for the EPR is clear. I was unsure which new result for the stationary distribution the authors were referring to, since various such results appear in Refs. 14-17.

  2. The parameter \alpha in Eq. 4 seems a little redundant, as it is referred to only once.

  3. The authors' self-consistent derivation of the Fourier law is convincing. I wonder if they could comment on its relation with the other approaches referred to e.g. in Ref. 16, and those in Ref. 17 and PRB 11 2164 that are mentioned in the review of Bonetto et al.

  4. The squared-log curve in Fig. 4 is a difficult to see on the plot, even in colour.

There were a small few typos: - balistic -> ballistic (abstract and elsewhere) - there seems to be a sign error in the defn of $F_{kh}$ below Eq. 22 - underamped -> underdamped (Fig 4 caption)

  • validity: -
  • significance: -
  • originality: -
  • clarity: -
  • formatting: -
  • grammar: -

Author:  Alberto Garilli  on 2020-08-03  [id 913]

(in reply to Report 1 on 2020-07-02)

We thank the reviewer for the nice report and the interesting considerations. We would like to answer every point one by one:

  1. Together with the expression for the EPR we also refer as a main technical result for the stationary distribution to the second-order Lyapunov equation Eq. 15 (Eq. 17 in the new version).
  2. The parameter $\alpha$ in Eq. 4 (Eq. 6 in the new version) is necessary since the constant $\kappa$ is dimensional (heat conductivity) we are considering an extended system in 3 dimensions (assuming the temperature varying only in one dimension), then $\alpha$ represents the area of a section orthogonal to the dimension in which we let the temperature vary. This is slightly different from what we obtain in the stochastic approach on the 1-dimensional chain of oscillators where such section $\alpha$ does not appear. If you refer to the fact that we also made use of the same symbol $\alpha$ as an index in the basis of the eigenstates of $A$ then it is sufficient to change symbol, even if we think it is not strictly necessary since there is no ambiguity.
  3. We added a comment (page 8 in the new version, after Eq. 27) in which we compare our model with the one in PRB 11 2164. The main difference is that the authors of that paper consider the equipartition to be satisfied by each oscillator, finding a linear temperature profile between the boundaries $T_0$ and $T_n$. In our work we considered instead the profile which minimizes the EPR, giving an exponential profile and reproducing the temperature frustration in the low damping limit. In this way it is possible to say that the best way to minimize EPR is to reduce the heat flow to the colder reservoirs.
  4. In the plot Fig. 4 the aim is to point out that the EPR scales with the number of oscillators as $1/n$. However, observing that the squared-log curves are not easy to see is definitely a good point and we we modified the figure in order to make all the remarkable informations visible.

Thank you for noticing the typos, the sign on the definition of the thermodynamic force had to be inverted.

---

## Round 1 · Referee Report · Anonymous (Referee 2) · 2020-7-7

Report

The authors study analytically and numerically the entropy production rate in a system connected to two reservoirs with different temperatures. The microscopic model in consideration is a set of classical oscillators coupled to dissipative environments. For this model a formula for the minimal rate of entropy is given. In the over-damped limit of the model the the minimal rate of entropy follows the one which can be derived from Fourier law. On the other hand, in the low-noise limit the formula predicts the ”equipartition frustration” which cannot be derived from Fourier law. The paper’s novelty is in the derivation of the minimal entropy production rate which is consistent with the one derived from Fourier law in the over-damped limit. This can be ultimately interpreted as self-consistent derivation of the Fourier law. The paper is clearly written with respect to the main points of the paper which are well explained. I can recommend for publication. However, before publication the authors should consider following minor comments:

•Above Eq. (3): The authors refer to ”the equation” without quoting the equation.
• Above Eqs. (5) in the discussions of the condition of non-oscillating ends - the condition $p_0$ = 0 is missing.
• The authors could add some reference for Ornstein-Uhlenbeck process and Kramers diffusion equation which might be helpful for non-specialized readers.
• Missing parenthesis in eq. (16).
• Since eq. (19) is one of the main result of the paper, more steps of the derivation of eq. (19) from eq. (18) should be provided.
• The discussion of discretized equivalents after eq. (24) should be improved. For example the connection of eqs. (23) and (A.2) is not clear to me.
• In appendix B: the reference to eq. (14) supposes to be a reference to eq.(18). The authors should double-check the cross-referencing of the equations.
  • validity: -
  • significance: -
  • originality: -
  • clarity: -
  • formatting: -
  • grammar: -

Author:  Alberto Garilli  on 2020-08-03  [id 914]

(in reply to Report 2 on 2020-07-07)

We thank the reviewer for the nice report and the interesting considerations. We would like to answer your points one by one:

  1. The equation we refer to is the Laplace equation $\nabla \log T^* = 0$ written above, inline. Since the equation is not labelled, we simply wrote “the Laplace equation” instead of “the equation” to make it more clear.
  2. The condition $p_0 = 0$ has been added.
  3. We already had the book “The Fokker-planck equation” by Risken in the bibliography. We also added “Stochastic processes in Physics and Chemistry” by Van Kampen.
  4. The missing parenthesis in Eq. (16) (Eq. 18 in the new version) has been added.
  5. We added a few more words in the derivation of Eq. 19 from Eq.18 (Eq. 21 from Eq. 20 in the new version), we hope now can result more clear to the reader.
  6. Some lines have been added in the discussion about discretized equivalents (page 7 after Eq. 26 in the new version), we hope now can result more clear to the reader.
  7. The references in the appendices have been fixed, we noticed that some labelling on equation were missing in the LaTeX source.

---

## Round 2 · Referee Report · Anonymous (Referee 1) · 2020-8-4

Report

I am satisfied with the authors' changes and recommend the manuscript for publication.

(please note a small typo on p9, kind -> kind of)

---

## Round 2 · Referee Report · Anonymous (Referee 2) · 2020-8-13

Report

The authors gave satisfactory answers to my comments. I recommend the manuscript for publication.

---

## Round 2 · List of Changes

• We added some additional references [15,23,29]
  • The typos have been corrected. In particular the definition of the thermodynamic forces $F_{k h}$ under eq. 26
  • Some further explanations on some passages have been added and labelled in red
  • We corrected some labelling in the main text and in the appendices
  • We changed fig. 4 with a better picture so that it is clearer than before
  • We added the condition $p_0$ = 0 among the conditions of non-oscillating endpoints
  • We added a missing parenthesis in eq. 18

---

## Editorial Decision

published